# STABILITY AND GENERALISATION IN BATCH RL

## ABSTRACT

Overfitting has been recently acknowledged as a key limiting factor in the capabilities of reinforcement learning algorithms, despite little theoretical characterisation. We provide a theoretical examination of overfitting in the context of batch reinforcement learning, through the fundamental relationship between algorithmic stability (Bousquet & Elisseeff, 2002)–which characterises the effect of a change at a single data point–and the generalisation gap–which quantifies overfitting. Examining a popular fitted policy evaluation method with linear value function approximation, we characterise the dynamics of overfitting in the RL context. We provide finite sample, finite time, polynomial bounds on the generalisation gap in RL. In addition, our approach applies to a class of algorithms which only partially fit to temporal difference errors, as is common in deep RL, rather than perfectly optimising at each step. As such, our results characterise an unexplored bias-variance trade-off in the frequency of target network updates. To do so, our work extends the stochastic gradient-based approach of Hardt et al. (2016) to the iterative methods more common in RL. We find that under regimes where learning requires few iterations, the expected temporal difference error over the dataset is representative of the true performance on the MDP, indicating that, as is the case in supervised learning, good generalisation in RL can be ensured through the use of algorithms that learn quickly.

## 1 INTRODUCTION

A central aim of machine learning theory is to provide wort-case bounds on overfitting. However, in reinforcement learning (RL), which typically employs online learning on an unlimited stream of data and does not minimise an empirical risk, generalisation is more difficult to characterise than in the supervised learning case. This said, overfitting is still observed in RL settings, with various definitions of data scarcity. This work looks to characterise generalisation and overfitting in *batch, off-policy* RL. Batch RL has been of particular recent interest, for reasons of data efficiency, policy safety, or as a component of a more complex online algorithm. Despite this, overfitting has never been directly characterised in this context. Fortunately, batch RL makes use of a limited data regime close to the ERM formulation, allowing us to contribute such an analysis.

Algorithmic stability (Bousquet & Elisseeff, 2002) describes the change in the model learned by an algorithm when the data employed is changed at a single point. It is a useful property, as it bounds the generalisation gap–the difference between error on our dataset, and the true population error. As such, if we can bound the stability of a batch RL algorithm, we can also make guarantees about how prone it is to overfitting: how well the error on our dataset represents the error across the whole MDP.

Specifically, this work formulates an off-distribution notion of *uniform stability* (Bousquet & Elisseeff, 2002), suitable for batch policy evaluation algorithms. We show the relationship between this variant of stability and an off-policy generalisation gap. This allows us to characterise overfitting in batch RL via stability, answering the question: *when can we expect a batch RL algorithm to generalise well?* This framework is then applied to a partially fitted variant of temporal difference (TD) learning, that bridges the gap between fitted and traditional TD methods. Importantly, no *realisability* assumptions about the ability of the function class to fit values are needed.

The iterative, partially fitted batch method employed here means that our result characterises the use of target networks (Mnih et al., 2015), common in deep RL algorithms. Our bound expresses a bias-variance trade-off in the number of steps taken between updates of the target network–as more

steps are taken, less data is incorporated into the update dynamics, leading to a higher variance, but lower bias procedure, as the value targets are also less correlated with the value parameters.

## 2 RELATED WORK

Our work builds directly on the results of Hardt et al. (2016), which establishes stability bounds for stochastic gradient methods in non-convex optimisation settings. That work in turn builds on *algorithmic stability*, a topic in learning theory that characterises the generalisation gap in a manner agnostic to model capacity (Rogers & Wagner, 1978; Devroye & Wagner, 1979; Bousquet & Elisseeff, 2002).

Bounds for batch RL have seen significant recent interest, especially for linearly parameterised value estimators. Existing bounds in this setting study generalisation by directly bounding the suboptimality of value function estimators, rather than the generalisation gap. Most relevant here is the work of Duan & Wang (2020), where tight upper bounds on finite time regret are given for linear value function approximators applied to fitted off-policy evaluation, but restricted to the finite-horizon RL problem, where data can be separated into specific trajectories, and under a function approximation class that is closed under Bellman backups. Fan et al. (2020) provides generalisation bounds in deep RL with use of a target network but these are capacity dependent and do not characterise partial fitting. Other key bounds for batch fitted methods include those of Murphy (2005); Antos et al. (2007); Munos & Szepesvári (2008); Tosatto et al. (2017), though settings and assumptions differ from those used here. Zhang et al. (2021) also characterise the convergence properties of a target network used with linear function approximation.

Furthermore, exponential worst-case lower bounds have been constructed for batch RL. Chen & Jiang (2019) show that concentration assumptions are needed for polynomial-time learning without strong assumptions on the function approximation class. Zanette (2021); Wang et al. (2020); Amortila et al. (2020) provide exponential lower bounds for batch RL with only realizability and coverage assumptions. Wang et al. (2021) investigate these lower bounds empirically for fitted $Q$-iteration, and provides upper bounds using either realizability, or strong concentration assumptions.

Direct characterisation of overfitting in RL has seen more interest recently, though the vast majority of work thus far is empirical. To our knowlege, Wang et al. (2019) provide the only theoretical characterisation of the generalisation gap beyond this work, though it applies in the online setting, depends on the model complexity, and makes the restrictive assumption that the reparameterization trick (Kingma & Welling, 2013) can be applied to the random variables of the MDP. Francois-Lavet et al. (2019) characterise overfitting asymptotically for batch RL as applied to partially observable Markov decision processes with finite observation, state, and action spaces. Empirical characterisation of overfitting in RL across several settings can be found in Zhang et al. (2018a), Zhang et al. (2018b); Cobbe et al. (2019), and Packer et al. (2019).

## 3 PRELIMINARIES

We consider the infinite horizon discounted RL setting. The agent interacts with an environment, formalised as a Markov Decision Process, $\mathcal{M}$ with (potentially infinite) state space $\mathcal{S}$, a finite action space $\mathcal{A}$, transition kernel $P : \mathcal{S} \times \mathcal{A} \times \mathcal{S} \rightarrow [0, 1]$, bounded stochastic reward kernel $R : \mathcal{S} \times \mathcal{A} \rightarrow \mathcal{P}(-r_{max}, r_{max})$, where $\mathcal{P}(a, b)$ is a probability distribution with bounded support on the interval $[a, b]$ and scalar discount factor $\gamma \in [0, 1)$. The agent's behaviour is determined by a policy that maps a state to a distribution over actions: $pi : \mathcal{S} \times \mathcal{A} \rightarrow [0, 1]$. We seek to optimise (in the control case), or estimate (in the policy evaluation case) the expected discounted sum of future rewards starting from a given state. This quantity is given by the state value function, $V(S) = \sum_{A \in \mathcal{A}} \pi(S, A) Q(S, A)$, with $Q : \mathcal{S} \times \mathcal{A} \rightarrow [-r_{max}/(1 - \gamma), r_{max}/(1 - \gamma)]$, the action value function, given recursively through the Bellman equation:

$$Q(S, A) = \bar{r}(S, A) + \gamma \mathbb{E}_{S' \sim P(S,A), A' \sim \pi(S')} \left[ Q(S', A') \right], \tag{1}$$

where $\bar{r}(S, A)$ is the mean reward as sampled from the reward kernel. The Bellman operator $\mathcal{T}^{\pi}$ projects functions forwards by one step through the dynamics of the MDP:

$$\mathcal{T}^{\pi}(Q)(S, A) = \bar{r}(S, A) + \gamma \mathbb{E}_{S' \sim P(S,A)} \left[ \mathbb{E}_{A' \sim \pi(S')} \left[ Q(S', A') \right] \right]. \tag{2}$$

$\mathcal{T}$ is a $\gamma$-contractive mapping and thus has a fixed point. The fixed point of this operator corresponds to the true value of the policy in question (Puterman, 2014). We make use of a feature mapping to encode states and actions, $\phi : \mathcal{S} \times \mathcal{A} :\to \mathbb{R}^d$. When estimating MDP values, we employ parameterised value functions with a linear approximation: $Q_w(\phi(S,A)) = \phi^\top(S,A)w$.

## 3.1 BATCH RL

In batch RL, rather than interacting directly with the MDP, the algorithm is given a fixed dataset, with samples drawn from the transition and reward kernels of the MDP. The agent uses these samples to estimate either the value of a fixed *target policy* $\pi$, or to learn an optimal policy, $\pi_*$. Usually, the state-action distribution induced by the *target policy* differs from that of the dataset, so batch RL is generally viewed as an extreme case of off-policy RL, in which additional data cannot be generated from any policy. In this paper, we focus on off-policy evaluation, leaving control to future work.

Fitted temporal difference methods are a class of batch RL algorithms that iteratively fix a *target estimator* for the value function, $g : \mathcal{S} \times \mathcal{A} \to \mathbb{R}$, and then minimise the mean squared empirical Bellman error (MSBE) of a separate estimator, $f_w : \mathcal{S} \times \mathcal{A} \to \mathbb{R}$, over a batch of data:

$$\frac{1}{D} \sum_{i=1}^{D} \min_w (f_w(s_i, a_i) - \hat{\mathcal{T}}g(s_i, a_i))^2, \tag{3}$$

where $\hat{\mathcal{T}}$ is the pointwise empirical Bellman operator:

$$\hat{\mathcal{T}}^\pi(Q)(S,A) = r + \gamma \sum_{A'} \pi(A'|S')Q(S',A'), \tag{4}$$

and $r$ and $S'$ are sampled by taking action $A$ in state $S$. Once the minimum is attained, $g$ is updated to match $f_w$. This procedure repeats until convergence.

In practice, particularly when using neural networks, the minimiser of (3) cannot be found in closed form. Furthermore, it may be too computationally expensive, or undesirable from the perspective of learning dynamics, to reach this optimum. Instead, partial optimisation is performed: a few updates of an iterative optimisation procedure often suffice. We summarise this process using linear function approximation and expected SARSA as our off-policy estimator of the Bellman error in Algorithm 1.

---

**Algorithm 1** Partially Fitted Linear Expected SARSA

---

**Input:** $D \sim \mathcal{D}_\mu$, learning rate $\alpha$, target policy $\pi$
**Output:** value parameters $w$
  1: initialize value parameters $w_0$
  2: **for** epoch $i = 0, ..., N$ **do**
  3:     fix $\bar{w} = w_{iK}$
  4:     sample minibatch of size $K$, $U = \{(S_j, A_j, R_j, S'_j)\}$
  5:     **for** update $k = 1, ..., K$ **do**
  6:       $t = iK + k$
  7:       $\delta_u(w) = \left(\phi(S_k, A_k)^\top w - R_k - \gamma \sum_{A'} \pi(S'_k, A')\phi(S'_k, A')^\top \bar{w}\right)$
  8:       $w_t = w_{t-1} + \alpha\phi(S_u, A_u)\delta_u(w_{t-1})$
  9:     **end for**
10:     project $w = \text{argmin}_{||w'|| \leq w_{max}} ||w - w'||$
11: **end for**
12: **return** $w_t$

---

Algorithm 1 also includes a projection step for the purposes of the theory in Section 4. After each epoch $i$, the value parameters, $w$ are projected to a closed ball of fixed radius. This prevents the step sizes from growing too quickly, as our update rule is non-Lipschitz for unbounded parameter spaces.

By varying the hyperparameters of Algorithm 1, we move between traditional and fitted temporal difference algorithms. If the minimiser of (3) exists, setting $K$ large enough to allow for convergence recovers fitted expected SARSA, while setting $K = 1$ recovers expected SARSA applied to the uniform distribution over data.

**Update Rules and Parameterisation**   One challenge that arises when treating the stability of Algorithm 1 comes from the fact that we need to reason over the behaviour of two separate sets of parameters–both the value and target parameters affect the update rule. In order to manage this, we propose a unified parameterisation for which we maintain a single set of parameters, $\hat{w}$ which combines the value parameters and the frozen ones:

$$\hat{w} := \begin{bmatrix} w \\ \bar{w} \end{bmatrix}$$

With this setup we need to use two update rules that operate on the same set of parameters. The parameter fitting update is given by:

$$G(\hat{w})_{s,a,r,s'} := \hat{w} - \alpha \begin{bmatrix} \phi(s,a) \\ \mathbf{0} \end{bmatrix} \left( \phi(s,a)^\top w - r - \gamma \sum_{a'} \pi(a'|s')\phi(s',a')^\top \bar{w} \right) \quad (5)$$

Where $\mathbf{0}$ is the zero vector in $d$ dimensions. This is identical to performing the fitted TD update on $w$ and does not alter the value of $\bar{w}$. When we need to update the frozen parameters, we simply use an assignment:

$$\hat{G}(\hat{w}) = \begin{bmatrix} w \\ w \end{bmatrix} \quad (6)$$

This setup allows our update rules to depend only on the data being used to perform that update, rather than having an implicit dependency on previous updates, due to the fact that there are two sets of parameters being maintained.

## 3.2   ALGORITHMIC STABILITY

While most learning theory characterises generalisation performance as a function of the complexity of the model class, algorithmic stability (Bousquet & Elisseeff, 2002) uses the statistical properties of the learning algorithm itself to provide bounds on generalisation.

Let $\Omega$ be the space of all data points, and let $\mathcal{W}$ be the class of models that we are learning over. Let $\mathcal{D}$ be a probability distribution over $\Omega$. In general, a (stochastic) machine learning algorithm $\mathcal{A}$ maps a sampled dataset of size $n$, $D = X_1, ..., X_n : X_i \sim \mathcal{D}$ to a model, $w \in \mathcal{W}$ with the aim of minimising a scalar *loss* function (risk), $l : \mathcal{W} \times \Omega \to \mathbb{R}$, over the population:

$$\mathcal{R}(w)_\mathcal{D} = \mathbb{E}_{x \sim \mathcal{D}}[l(w; x)]. \quad (7)$$

Generally, we only have access to the data in the dataset, and thus cannot optimize (7) directly; instead we can optimise the empirical counterpart:

$$\mathcal{R}_{emp}(\mathcal{A}(D)) = \frac{1}{|D|} \sum_{x \in D} l(\mathcal{A}(D); x). \quad (8)$$

However, (8) can be optimised by memorising $D$ and guessing randomly for other data, leading to *overfitting* and poor performance of $\mathcal{A}(D)$ on (7). An algorithm overfits if (8) is low while (7) is high, i.e., it has a high *expected generalisation gap*:

$$\mathcal{G}(\mathcal{A})_\mathcal{D} = \mathbb{E}_{D \sim \mathcal{D}, \mathcal{A}} \left[ \mathcal{R}_{emp}(\mathcal{A}(D)) - \mathcal{R}(\mathcal{A}(D)))_\mathcal{D} \right]. \quad (9)$$

Small $\mathcal{G}(\mathcal{A})_\mathcal{D}$ implies that minimising $\mathcal{R}_{emp}$ is a reasonable way to minimise $\mathcal{R}_\mathcal{D}$.

In supervised learning, there is an intimate connection between the generalization gap and *stability* (Devroye & Wagner, 1979), the sensitivity of an algorithm to small perturbations in the data set. Intuitively, suppose we fix a learning algorithm's randomness, then apply it to two datasets that differ at exactly one point. If the algorithm outputs two similar models that yield similar loss, then the algorithm is stable and generalises well. While many forms of stability have been investigated, the simplest and most popular is *uniform stability* (Bousquet & Elisseeff, 2002). An algorithm is $\epsilon$-uniformly stable if it satisfies:

$$\sup_{z,x,x',D} \mathbb{E}_\mathcal{A} |l(\mathcal{A}(D \cup x), z) - l(\mathcal{A}(D \cup x'), z)| \le \epsilon,$$

where, $z$, $x$, and $x'$ are all data points. The expectation is taken over algorithmic randomness. The idea is to bound the worst case difference in losses if we apply the same algorithm to datasets that differ at exactly one point. In supervised learning, the expected generalization gap of an algorithm is bounded by its stability, i.e., if $\mathcal{A}$ is $\epsilon$-uniformly stable, then $G(\mathcal{A})_{\mathcal{D}} \leq \epsilon$ (Bousquet & Elisseeff, 2002).

## 4 STABILITY AND GENERALISATION IN BATCH RL

In this section, we characterise the propensity of Algorithm 1 for overfitting. To do so, first we develop a novel off-distribution notion of uniform stability. Then, we characterise the deviation of our model parameters (and thus the loss) when Algorithm 1 is applied to a perturbed dataset, as opposed to the original data. To do so, we bound this gap recursively as is done by Hardt et al. (2016), with key adaptations to handle correlated updates and nonstationary objectives. As more parameter updates of the algorithm are performed, the effects of the changed data point compound and its stability declines.

To bound the distance between our two sets of parameters, we introduce some relevant assumptions:

**Assumption 1** $\phi$ *maps to a compact set in* $\mathbb{R}^d$, *and is uniformly bounded, with* $||\phi|| \leq \phi_{max}$.

The assumption of bounded features is common in the literature (Munos & Szepesvári, 2008) and in general can be satisfied by scaling features without loss of generality.

### 4.1 OFF-DISTRIBUTION ALGORITHMIC STABILITY

Due to the off-policy nature of batch RL, the notion of uniform stability cannot be directly applied. While we fit our value function over a fixed data distribution, we are interested in the loss over the target distribution $\mathcal{D}'$. Formally, we want to optimise:

$$\mathcal{R}(\mathcal{A}(D))_{\mathcal{D}'}, \ \ D \sim \mathcal{D}. \tag{10}$$

Therefore, we employ an off-distribution generalisation gap, which compares the expected empirical regret to the population regret of the output model with respect to a distribution other that from which the training data was sampled:

$$\mathcal{G}(\mathcal{A}, \mathcal{D}, \mathcal{D}') = \mathbb{E}_{\mathcal{A}, D \sim \mathcal{D}} \left[ \mathcal{R}_{emp}(\mathcal{A}(D)) - \mathcal{R}(\mathcal{A}(D))_{\mathcal{D}'} \right] \tag{11}$$

This quantity characterizes how overfit our algorithm is to our dataset, relative to our expected performance on the *target* data distribution.

In order to bound the off-distribution generalisation gap, we introduce a new notion of stability:

**Definition 1 (Uniform Off-Distribution Stability)** *An algorithm* $\mathcal{A}$ *is* $\epsilon$-*uniformly off-distributionally stable with source* $\mathcal{D}$ *and target* $\mathcal{D}'$ *distributions, if* $\mathcal{D}$ *and* $\mathcal{D}'$ *share support and:*

$$\sup_{z, x, x', D} \left| \mathbb{E}_{\mathcal{A}} \left[ l(\mathcal{A}(D \cup x), z) - \frac{P_{\mathcal{D}'}(z)}{P_{\mathcal{D}}(z)} l(\mathcal{A}(D \cup x'), z) \right] \right| \leq \epsilon \tag{12}$$

This definition is similar to uniform stability but one of the losses is corrected with the ratio of the probabilities of the test point, under target and source distributions. While importance sampling methods appear frequently in RL (Precup, 2000), this ratio particularly resembles the one treated in marginalised importance sampling methods (Xie et al., 2019), which perform policy evaluation by directly estimating this ratio, where $\mathcal{D}$ is the (potentially mixed) data distribution, and $\mathcal{D}'$ is the steady state distribution induced by the target policy.

Crucially, the relationship between uniform off-distribution stability and the off-distribution generalisation gap is identical to that between uniform stability and the on-distribution generalisation gap:

**Lemma 1** *If an algorithm* $\mathcal{A}$ *is* $\epsilon$-*uniformly off-distribution stable for distributions* $\mathcal{D}$ *and* $\mathcal{D}'$ *with shared support, then:*

$$\mathcal{G}(\mathcal{A}, \mathcal{D}, \mathcal{D}') \leq \epsilon.$$

The proof of Lemma 1, found in the appendix, closely follows the proof for the supervised setting, with the probability ratio introduced to correct for distribution shift. However, this ratio is problematic, as it can grow large if a likely event under $\mathcal{D}'$ is unlikely under $\mathcal{D}$. While it does not depend on a product of ratios like return-based importance sampling methods do (Munos et al., 2016), it can grow unboundedly. As such, we make an additional assumption bounding the ratio.

**Assumption 2** *The density ratio in (12) is bounded from above:*

$$\sup_z \frac{P_{\mathcal{D}'}(z)}{P_{\mathcal{D}}(z)} \leq \rho_{max}.$$

This is the weakest form of a "low distributional shift" assumption that is common in the literature (Munos & Szepesvári, 2008; Fan et al., 2020; Wang et al., 2019). Chen & Jiang (2019) demonstrate that this assumption is necessary for polynomial learning guarantees absent strict assumptions on the function approximation class. In our context, it ensures that our choice of distributions does not already ensure overfitting, by concentrating the data that we are learning on in regions of the MDP that are not relevant to the target distribution.

## 4.2 LINEAR PARTIALLY FITTED EXPECTED SARSA IS $\eta$-EXPANSIVE

Given the smoothness of our linear function approximators, to bound the uniform off-distribution stability and thus bound the generalisation gap, we only need to bound the difference between the parameters of two models encountering datasets that differ at one point. Since our algorithm is iterative we can bound this gap recursively. This gives rise to one of two cases: if the data points encountered by the two models are different, the losses are different and the parameters diverge; if the data encountered by our two models is the same, the update still may increase the distance between their parameters, since the initial parameters differ. We defer bounding of the first case divergence to Section 4.3. In this section, we bound the second case using the notion of $\eta$-expansivity. An update rule, $G : \Theta \rightarrow \Theta$ is $\eta$-expansive if:

$$\sup_{v,u \in \mathcal{W}} \frac{||G(v) - G(u)||}{||v - u||} \leq \eta < \infty.$$

The following lemma describes the expansivity of the TD update rule given by (5) in the inner loop of Algorithm 1. It will be applied whenever an update is performed on data that does not correspond to the perturbed point.

**Lemma 2** *Under Assumption 1, and the additional projection step, the update rule $G(\hat{w})_X$ is $\eta$-expansive, with:*

$$\eta \leq 1 + \alpha \left(1 + \gamma\right) \phi_{max}^2. \tag{13}$$

The proof, found in the appendix, follows directly from the definition of expansivity. Our choice of parameterisation allows the update to depend only on the current data point, rather than all previous data points in the epoch. It means that once the parameters of our two algorithms are no longer the same, we can still provide a tight bound on how much further they become on subsequent updates that use the same data.

## 4.3 LINEAR PARTIALLY FITTED EXPECTED SARSA IS $\sigma$-BOUNDED

When the two algorithm rollouts encounter the data point that differs between them, the update rules used for each model is different, so we cannot use $\eta$-expansivity to govern the parameter gap. In this situation, we employ the worst case gap, which simply limits the size of steps in parameter space. An update rule is $\sigma$-bounded if:

$$\sup_{w \in \mathcal{W}} ||G(w) - w|| \leq \sigma.$$

Thanks to the projection step, updates cannot grow without bound. However, the magnitude of the update can depend exponentially on the number of steps taken since the last projection.

**Lemma 3** *Under Assumption 1, for finite integers $k$ and $p$, and the additional projection step, the update rule $G(w)_{X_k}$ is $\sigma$-bounded, with:*

$$\sigma \leq \alpha\phi_{max}\left((1+\gamma)\phi_{max}w_{max} + r_{max}\right)\left(\alpha\phi_{max}^2 + 1\right)^k. \tag{14}$$

The proof is found in the appendix. While this bound is highly pessimistic, and thus exponential in $k$, in practice algorithms typically employ only a few updates between batches. This acts as a Lipschitz-style constant, uniformly bounding the size of the parameter update step. This is possible because at the start of every epoch, the parameters are bounded by $||w|| \leq w_{max}$. When $K$ is finite, since only a finite number of finitely sized steps are taken, there exists a second ball beyond which the parameters never reach. Since feature vectors are bounded and the TD error is finite for bounded $R$, $W$, and $\phi$, we can compute this effective upper limit on step sizes. Because the parameters cannot take arbitrarily big steps, when the algorithm (infrequently) encounters the data point that has been changed, we can provide a limit on how far the parameters of the algorithm will move.

The dependence on $K$ here suggests that, when the target network parameters are updated infrequently, step sizes can increase. Later we will see that this increase in step size makes the algorithm more sensitive to the data encountered, potentially leading to an increase in overfitting.

### 4.4 LINEAR PARTIALLY FITTED EXPECTED SARSA IS UNIFORMLY STABLE

Using the boundedness and expansivity properties of our update rule, we proceed to our main result:

**Theorem 1** *Under Assumptions 1 (under supremum norm) and 2 and the projection step applied under supremum norm, for monotonically decreasing $\alpha_t \leq \frac{c}{t}$, after $T = NK$ updates, Algorithm 1 is $\epsilon$-uniformly off-distributionally stable, with:*

$$\epsilon \leq (1 - \rho_{max})((1+\gamma)d\phi_{max}w_{max} + r_{max})^2 + \rho_{max}E, \tag{15}$$

*where:*

$$E = \left((1+\gamma)d\phi_{max}w_{max} + r_{max}\right)^{\frac{2\beta c}{\beta c+1}} \left(\frac{1 + \frac{1}{\beta c}}{|D| - 1}\right) \left(2M\sqrt{d}Lc\right)^{\frac{1}{\beta c+1}} T^{\frac{\beta c}{\beta c+1}}$$

*and:*

$$M = \left(d^{\frac{3}{2}}(1+\gamma)^2\phi_{max}^2w_{max} + \sqrt{d}2(1+\gamma)\phi_{max}r_{max}\right)$$

*and:*

$$L = \phi_{max}\left((1+\gamma)\phi_{max}w_{max} + r_{max}\right)\left(\alpha\phi_{max}^2 + 1\right)^K,$$

*and:*

$$\beta = (1+\gamma)\phi_{max}^2.$$

The full proof of Theorem 1 can be found in the appendix, though we sketch relevant details here. First, by the effective boundedness of both our loss function and the ratio $\rho$, we have $(l_1 - \rho l_2) = (1 - \rho)l_1 + \rho(l_1 - l_2) \leq \infty$. Noting that this primarily depends on the gap between our (uncorrected) loss functions. As such we proceed by bounding the expected gap between the squared Bellman errors of our two parallel algorithms. We use a hitting time argument, which suggests the perturbed data point is unlikely to be encountered early, so the sets of parameters will not diverge until the step size has significantly decayed. This gives rise to a key lemma in the proof of Theorem 1:

**Lemma 4** *Let $w_t$ and $w'_t$ be the output from running Algorithm 1 on our original dataset, and the perturbed dataset respectively, with algorithm randomness held constant across both runs. Let $\zeta_t$ represent the parameter gap in supremum norm at time $t$: $|w_t - w'_t|_\infty$. Then, under Assumption 1 with supremum norm, and using supremum norm for the projection step, for every $t_0 \in \{1, ..., |D|)\}, z \in \Omega$, we have:*

$$\mathbb{E}_\mathcal{A}|MSBE(w_T; z) - MSBE(w'_T; z)| \leq \frac{2t_0}{|D|}((1+\gamma)d\phi_{max}w_{max} + r_{max})^2 + M\sqrt{d}\mathbb{E}[\zeta_T | \zeta_{t_0} = 0]. \tag{16}$$

This lemma is similar to Lemma 3.11 of Hardt et al. (2016), with several modifications for our setting. The first term is the worst-case loss gap, representing the situation where the perturbed point is encountered before $t_0$. The second term gives the loss gap at time $T$ in the event that the parameters have not yet diverged by time $t_0$, as the data encountered by the two instances of the algorithm has been the same up to that point.

The use of the supremum norm is unconventional here. We do this because under the supremum norm, the target network update is nonexpansive. While this leads to dependence on the dimension of the state space, it allows us to manage what would otherwise become a quickly intractable dependence on data previously encountered in the epoch.

Once Lemma 4 is established, all that remains is to bound the parameter gap, which we can do using our expansivity and boundedness rules from above. These rules allow us to recursively bound the gap at time $T$ as a function of $t_0$, using the expansivity rule for when each set of parameters encounters the same data, and the boundedness rule for when the parameter sets encounter the perturbed data point. We then minimize for $t_0$, and the proof is complete.

This result implies that Algorithm 1 does not overfit arbitrarily, despite the non-convexity of the overall optimisation problem. While our bound may become loose beyond the trivial bound for high $T$, for certain parameter regimes, and after few time steps, we can provide meaningful guarantees on generalisation, allowing us to ensure that our performance on the MDP is not too much worse than the performance on our batch of data. This means that under certain conditions, we can be confident that our policy evaluation is accurate.

**Bias-variance tradeoff of target network.** Another key insight here is the bias-variance tradeoff that our result demonstrates in $K$. While target networks are common in practice, they have never been analysed in the partial fitting setting used here, which allows us to compare the use of the target network with traditional TD methods. It is well-known (see, e.g. Fan et al. (2020)) that application of (5) with $K = 1$ does not correspond with optimisation of the MSBE, but rather a biased version, given by:

$$\mathbb{E}_z\left[MSBE(w; z)\right] + \gamma^2 \mathbb{E}_{s,a \sim \mu}\left[\mathbb{V}_{s',a' \sim \pi}\left[\phi(s', a')^\top w\right]\right] \tag{17}$$

While if $K$ is taken to infinity, we recover the "minimax" formulation for fitted policy evaluation, which does optimise the MSBE (Zhang et al., 2021). As such, we can interpret the exponential dependence of our bound on $K$ as describing a bias-variance tradeoff, where for low $K$, we optimise a biased objective but with less variance and thus increased stability. On the other hand, for large $K$, step sizes increase, and thus so does the dependency on the data, but we recover an unbiased objective.

While our bound provides unique characterisation of overfitting in batch RL, it suffers limitations in its applicability as well. As a bound in expectation, it is significantly weaker than a high-probability bound. In this respect, it suffers from the same challenges that prevent application of concentration inequalities discussed by Hardt et al. (2016).

## 5 EMPIRICAL RESULTS

While the main goal of this paper is to establish the theoretical results presented above, we also present some empirical results in this section. In order to verify that our worst-case analysis holds true in practice, we examine the effects of varying the parameters of Fitted Linear Expected SARSA (Alg.1) on a noisy variant of the episodic Boyan chain environment (Boyan, 2002).

The first state in the chain is given by the axis-aligned unit vector $[1, 0, 0, 0]$. Advancing along the chain involves interpolation between subsequent single-component vectors, with the next component increasing by $0.25$ for each state advanced, and decreasing the previous by the same. The agent can either move one state ahead or two, save for the second to last state, in which the agent must advance one. In our variant, in order to examine the effects of overfitting, we add mean-zero noise to both the transition and reward functions. Centered Gaussian noise is added to the rewards and uniform noise in $(-0.125, 0.125)$ to each state. We use bounded uniform noise to prevent state aliasing and thus maintain the Markov property. In order to allow for multiple solutions and more possibility of overfitting, we augment the state space with polynomial features up to degree four. Our target policy selects uniformly between moving ahead one state and moving ahead two states, while the dataset

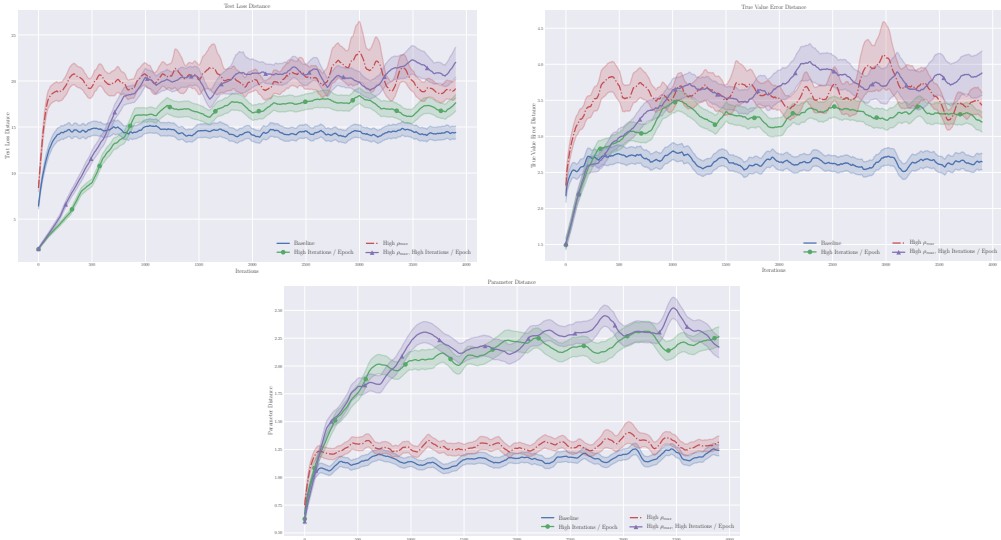

Figure 1: Results of empirical investigation of stability. Increasing the number of iterations per epoch and increasing the divergence between the sampling and the target distribution significantly decrease stability and thus generalisation.

undersamples states where the first component is nonzero. This is an extreme case, as the sampling distribution does not even correspond to a policy, since the initial state is undersampled as well.

The most important parameters in our bound are the number of steps taken between epochs, $k$, and upper bound of the off-policy coefficient, $\rho_{max}$, which trades off between our vacuous upper bound, and the learning-dependent one. Running fitted linear expected SARSA on two identical datasets with one point changed between rollouts, and randomness held constant, we investigate the effects of varying these parameters on various empirical proxies for the stability: namely the parameter gap ($|w - w'|$), the Bellman error gap on a held out test set sampled from the target policy ($\mathbb{E}_z |MSBE(w; z) - MSBE(w'; z)|$), and the gap between statewise true errors, where ($|TE(w) - TE(w')|$), where

$$TE(w) := \sqrt{\sum_{S,A} \left(\phi(s, a)^\top w - Q_\pi(s, a)\right)^2},$$

and the sum is taken over the true states, without noise added. While these are all computed in expectation rather than supremum, they give indications as to how varying relevant parameters can influence the stability of the algorithm. Each configuration is run for $5000$ parameter updates. The results of the experiment appear in Fig. 1. Experiments were repeated over $500$ random seeds, with lines representing the mean gap and shaded areas give one standard error. We find that, as reflected in the bound's dependence on T, until convergence, across all runs, the parameter gap increases with the number of updates. Furthermore, we find that in correspondence with the bound's exponential depencence on $k$, increasing the number of iterations per epoch leads to an increase in overfitting, where divergence of parameters, MSBE, and TE are all higher than when fewer iterations are used. Finally, we find that increasing the degree to which the data is sampled off-policy leads similarly to an increase in overfitting, which though smaller in terms of parameter divergence, leads to comparable instability as the increased number of iterations.

## 6 CONCLUSIONS

We have provided a finite update bound on generalisation in the batch RL context for fitted TD methods. In doing so, we have established an off-distribution formulation of algorithmic stability and directly characterised overfitting in fitted TD methods. Furthermore, we have shed new light on a bias-variance tradeoff in the frequency of updates of the target network. In future, we look to find high-probability bounds rather than bounds in expectation, as well as investigate the generalisation properties of control algorithms.

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
