# OpenReview forum: "Stability and Generalisation in Batch Reinforcement Learning"
_ICLR.cc/2022/Conference — ICLR 2022 Submitted_

### Official Review · Reviewer_HjVF · 2021-10-29

**Correctness:** 3
**Technical Novelty And Significance:** 3
**Empirical Novelty And Significance:** 2
**Recommendation:** 3
**Confidence:** 4

**Details Of Ethics Concerns:**

I did not find any obvious ethical concerns.

**Main Review:**

PROS

I'd like to say that extending the stability-based analyses to bath RL settings is an interesting line of thought.

CONS

This paper is not sufficiently close to publishable form, in my opinion. This comment has to do with the amount of editorial changes that would need to be addressed (see my editorial feedback below), and also with concerns around definitions and mathematical formalism.

I found it difficult to follow the switch from RL settings to supervised learning settings, and how the latter helps the former. In particular, Section 4 focuses on definitions that, as far as I understand, are framed in the supervised learning context, while the intention is to carry them over to the setting ob batch RL. Perhaps this means that the way the paper is written isn't helping.

Perhaps the difficulty starts from Sections 3.1 and 3.2. Section 3.1 describes the batch RL setting and notations. Section 3.2 describes algorithmic stability and sets the definitions and notation, which are borrowed from the supervised learning setting I think. Perhaps what's missing is some connection between these two sections: That the loss function $l$ of Section 3.2 will be taken to be the squared loss so that the empirical risk defined in Eq. (8) corresponds to the empirical mean squared Bellman error defined in Eq. (3)?

In general it is unclear what content is from previous works (or even well-known) and what content is novel.

The previous is related to an overall lack of referencing, to attribute to the literature and to support claims.


EDITORIAL FEEDBACK

First line of abstract: delete "recently"

Suggested rewrite: "We provide a theoretical examination of overfitting in the context of batch reinforcement learning, through the fundamental relationship between algorithmic stability and the generalisation gap---the former characterises the effect changing at a single data point, and the latter can be used to quantify overfitting."

[No need for a reference here, it is better to give the reference in the introduction and related literature.]

Suggested rewrite: "To do so, our work extends the stability analysis of stochastic gradient methods given in previous work, to the iterative methods that are more common in RL"

[Again, no need for a reference here, which is more suitable to be given in the main body rather than the abstract.]

Replace "performance on the MDP" with "performance on the whole environment"

[Declare and use the acronym "MDP" in the introduction.]

Section 1 (Introduction):

I think the central aim of machine learning theory is to characterize learning. Studying overfitting might be one aspect at best. The claim that the aim is "worst-case bounds" may need to be supported, according to what is the intended message.

The sentence "overfitting is still observed in RL settings" calls for a reference.

So does the sentence "Batch RL has been of particular recent interest"

[Or at least insert something like "(see the related literature in Section 2 below)"]

Replace "close to the ERM formulation" with "close to the supervised learning setting"

Next paragraph: It is not clear what notion of "error" the authors have in mind. This is regarding the sentence that says "error on our dataset" and "error across the whole MDP" -- please clarify to help the readers.

Note that the meaning of "MDP" has not been declared.

Last line of page 1: Replace the dash with a colon.

Section 2 (Related Work):

"the results of Hardt et al. (2016), who established [...]"

Next paragraph: "linearly parametrised value function estimators" (insert "function")

"though their settings and assumptions differ from those used here." (insert "their")

Next paragraph, sentence about Wang et al. (2021): replace "provides" with "provide"

British vs. American spellings: choose one and use consistently throughout the paper.

[e.g. currently both "realisability" and "realizability" are used, and "characterises" and "characterizes" and may be others.]

Section 3 (Preliminaries):

"formalised as a Markov Decision Process (MDP), denoted $\mathcal{M}$, with [..]"

Insert a reference where the reader may see all the details of these definitions.

Replace "pi" with "$\pi$"

Suggested rewrite: "Typical problems seek to estimate (in the policy evaluation case) or optimise (in the control case) the expected discounted sum of future rewards [..]"

To make things clear I suggest to show explicitly the two expectation signs in the right hand side of Eq. (1). This means to show the expectation w.r.t. $S' \sim P(S,A)$, and separately the expectation w.r.t. $A' \sim \pi (S')$.

Line after Eq. (1): write the mathematical definition of $\bar{r}(S,A)$.

The expectations in Eq. (2) seem to be in the wrong order: First $S' \sim P(S,A)$ and next $A' \sim \pi (S')$, therefore the inner expectation should be w.r.t. $S' \sim P(S,A)$ and the outer expectation w.r.t. $A' \sim \pi (S')$. This order should be used also in Eq. (1).

Next paragraph: "The operator $\mathcal{T}^\pi$ is $\gamma$-contractive, and thus has a fixed point."

Next sentence: "The fixed point of $\mathcal{T}^\pi$ corresponds to the true value of the policy $\pi$ in question"

Last line of this paragraph: write "$\mathcal{Q}_w(\phi(S,A)) = w^\top \phi(S,A)$" or "$\mathcal{Q}_w(\phi(S,A)) = \phi(S,A)^\top w$"

Section 3.1:

Replace "to estimate either" with "either to estimate"

Eq. (3): The minimum should be outside the sum, I think?

The sentence "In practice, particularly when using neural networks, the minimiser of (3) cannot be found in closed
form" should be followed by some suitable reference(s) to support it.

Ii is not clear why it may be "undesirable from the perspective of learning dynamics, to reach this optimum" -- clarify or elaborate.

A reference for "expected SARSA" might be relevant.

Replace "Bellman error in Algorithm 1." with "Bellman error. The procedure is described in Algorithm 1."

Next paragraph: "the value parameters $w$ are projected" (remove the comma after "parameters")

Next paragraph: "By varying the hyperparameters of Algorithm 1, namely [..], we move between" (and fill in what should be in [..])

Overall, it looks that this section is presenting things that are known, but references are missing.

Line after Eq. (5): "where [,,]" (not capitalized)

Section 3.2:

Second paragraph: "Let $\mathcal{X}$ be the space of all data points, and let $\mathcal{W}$ be the space of parameters corresponding to the class of models [..]"

"a sampled dataset of size $n$, say $D = (X_1, ..., X_n)$ where $X_i \sim \mathcal{D}$, to a model $w \in W$, with the aim of [..]"

"the aim of minimising the \emph{risk}, which is the \emph{expected loss} for a given loss function $l : \mathcal{W}\times\mathcal{X} \to \mathbb{R}$ over the population distribution $D$:"

Use "$\mathcal{R}_\mathcal{D} (w)$" instead of "$\mathcal{R}(w)_\mathcal{D}$"

The discussion after Eq. (8) is misleading. Note that minimizing the risk entails minimizing the empirical risk and the gap. This is because minimizing only the empirical risk could lead to overfitting (big gap), and minimizing only the gap could be done with constant functions which may not minimise the risk.

Eq. (9): might be better to write $\mathcal{G}_\mathcal{D}(\mathcal{A})$ or $\mathcal{G}(\mathcal{A},\mathcal{D})$ on the left hand side.

Next paragraph: Replace "the simplest and most popular is" with "we focus on the notion of"

Warning: what's written at the bottom of page 4 looks like stability in expectation, this is different from the "uniform stability" defined by Bousquet and Elisseeff (2002), see their Definition 6 (their Eq. (7)).

The paper "Sharper bounds on uniform stability" (Bousquet, Klochkov, Zhivotovskiy, 2020) is relevant and as its title says it gave the sharper generalization bounds cased on uniform stability (sharper than the 2002 paper).

Section 4:

In Assumption 1, the second part is redundant: If the range of $\phi$ is compact, then $\sup_{(s,a)} \Vert \phi(s,a) \Vert$ is finite. Then you can take $\phi_\mathrm{max} = \Vert \phi \Vert$ (the sup norm).

Section 4.1:

First paragraph: It could be helpful for the reader to point of what are the fixed distribution that generates the training data, and the target distribution? With the batch RL problem in mind, these distributions must come from the interaction of policies with an environment?

It is not obvious what's the connection between optimization of Eq. (10) and the batch RL problem.

Missed some motivation for Definition 1. Why does the ratio of probabilities appear? Technically it is obvious, it is a trivial change of variable. What I am asking is for some motivation for why to define this as the proposed novel stability notion. Also whether/how it is computable.

Section 4.2:

What's called "$\eta$-expansive" is more commonly called "$\eta$-Lipschitz"

Section 4.3:

The name "$\sigma$-bounded" is not very informative. What this condition tries to do is control the distance between $G(w)$ and $w$ uniformly for all $w$'s.

Section 4.4:

Paragraph after Theorem 1: The reader may wonder what are $l_1$ and $l_2$? In order to understand the sketch of the proof.


Section 5:

The axis labels are extremely small, this makes the plots very difficult to read.

Would it be possible to see specific examples of batch RL problems where early stopping is advisable and gives better results than long training times? This would strengthen the support in favor of the argument that fast training improves generalization.

**Summary Of The Paper:**

As far as I can see, this work tries to leverage the connection between stability and generalization studied in supervised learning settings, and tries to build a similar connection for batch RL settings. To reason about stability in a batch RL problem, this work proposes a modified definition of stability that takes into account the two different distributions involved: the distribution generated by the interaction between the environment and the policy that generated the data, and the distribution corresponding to the policy being evaluated. Then the main result of the paper (Theorem 1) proposes an upper bound on the stability for fitted expected SARSA (and linear value function approximation).

**Summary Of The Review:**

One big concern is the writing of the paper: I do not think it is nearly ready to publishable form, as it looks that a large amount of work needs to be done to get there. These deficiencies affect the paper's readability and quality.

I think the main idea of the paper might be interesting, and the results may have some value. I did not find any obvious flaws. The theoretical contribution is not groundbreaking, as the presented results largely re-use proofs of existing results, but nevertheless there is some merit in extending to a new setting. I am not so sure that the section on experiments adds much value.

My evaluation is based mainly on considering the amount of fixes that need to be done for this work to be acceptable for publication.

---

> ### Author Response · Authors · 2021-11-23
> **Review Response**
>
> Thanks for the exceptionally detailed review. We largely agree with the editorial feedback that you've provided, and much of it will be included in subsequent versions of the paper. In particular, we're very grateful for the reference to the work by Bousquet, Klochkov, Zhivotovskiy, (2020), the suggestion about discussion around Definition 1, and indication that work is needed to bridge section 3.1 and 3.2. We feel that these changes will make the paper much easier to follow. Thanks again.

---

### Official Review · Reviewer_P8Es · 2021-11-03

**Correctness:** 2
**Technical Novelty And Significance:** 2
**Empirical Novelty And Significance:** Not applicable
**Recommendation:** 3
**Confidence:** 4

**Details Of Ethics Concerns:**

Not applicable.

**Main Review:**

The paper is firstly poorly written and it makes it very difficult to understand the setting of the problem, nature of the algorithm and the result and identify the correctness or implications of the final results. More importantly, the authors discuss the results as 'first' results on overfitting stating that "overfitting has never been directly characterised in this context". This is highly misleading as there is a substantial literature on understanding the necessary and sufficient conditions on the sample complexity of batch RL under varying assumptions on the data distribution, approximation power and statistical complexity of the value function/model class and algorithmic schemes. Various important quantities and the relevant expressions, thereof, are not well defined or not defined altogether at various places. Please see below for more details.

**Problem setting**: The paper studies the batch RL problem of offline policy evaluation for a discounted MDP when the data has been collected using a distribution $\mathcal{D}$. The authors study a value function based approach where the value function class is a set of linear functions of the state action features $\phi(s,a)$. The algorithm (Partially fitted linear expected SARSA) can be summarised as follows:

1. The algorithm runs in epochs where the two sets of linear value functions are maintained: $\phi \cdot w$ and $\phi \cdot \bar w$ where the latter plays the role of a target function.
2. In each epoch, the agent makes updates to the parameter $w$ by using gradient updates over $K$ transition tuples and the loss function: $\left( f_w(s,a) - r - \gamma E_{a' \sim \pi}[f_{\bar w}(s',a')] \right)^2$.
3. Projects the updated parameters $w$ onto a bounded ball and in the next epoch updates the target network to $\bar w = w$.

**Technical issues:** Now, this is where the details start to get murky. This is clearly an iterative algorithm which is performing a version of sample fitted value iteration. The only difference here from the well studied fitted policy evaluation problem (FPE) (see Le et al, 2019) is that the target network for the next epoch is updated only after partially minimizing the Bellman error instead of the full update in the usual case. This is what the authors refer to as 'partially fitted'. However, then the authors simply view this as a stochastic gradient update to a fixed objective and go on to study the stability properties of this SG method. The technical tools and proof approach used here is basedf on the paper by Hardt et al (2016) which analyzed the stability and generalization properties of stochastic gradient method for convex, strongly convex and non-convex losses. The main loss function which the authors study, the MSBE quantity, is never mentioned in the main paper or in the appendix. It is not clear as to how the authors view this as a single optimization objective and therefore, it is not clear as to what 'generalization' or 'overfitting' aspect is studied in the paper. In my initial reading, because of the strong optimization connection, I had expected an analysis of a residual Bellman error based method which actually minimizes a fixed objective instead of the iterative template of FQE.

Since the authors clearly have two steps of updating the target network as well as the main Q function, how is it equivalent to minimizing a mean-squared Bellman error (MSBE). Further, in the empirical loss written in the paper in eq 3, the expectation of the loss will not be equivalent to the desired Bellman error anyway. This has to do with the excess variance term which the authors mention later in the text. Therefore, how is the theory of ERM directly applied here without having a direct sum of decomposable loss function to begin with?

In the update rules and parameterization paragraph, the authors discuss the update $G(w)$ in detail, but how does $\hat G(w)$ factor into the analysis? I'm assuming one cannot simply ignore that in the stability bounds. The changes in magnitude of parameter changes in $w$ are discussed in the analysis, but the target network update is completely ignored. Hence, it will be helpful to expand on the whole setup and the exact reduction to the stochastic loss minimization setting in detail and then discuss how the tools from stability and generalization of ERM can be used. Along the same lines, clarifying the complete details of where smoothness is assumed (start of section 4.2), random permutation vs random sampling protocol for training, effect of K on the magnitude of parameter updates among other things seems necessary.

The above mismatch is evidently clear at many places where the authors simply lift the interpretation of results from Hardt et al in their setting. For instance, the authors use the following statement in their discussion in section 4.4: "ms. We use a hitting time argument, which suggests the perturbed data point is unlikely to be encountered early, so the sets of parameters will not diverge until the step size has significantly decayed". While the first part is true that for two datasets, the probability of the algorithm encountering the changed datapoint is proportional to $1/|D|$, the second does not make any sense in the paper's setting. For the Hardt paper, the discussion was in the context of SGM for non-convex losses with decreasing step sizes $\alpha_t = c/t$. However, for the iterative algorithm studied in this setting, having a decreasing sequence of step sizes can be counterproductive as the Bellman error for later target functions will not decrease and is antithetic to the underlying dynamic programming principle. Also, in the proof of Lemma 4, the last para says that $M$ comes from the definition of MSBE, which again is not defined anywhere.

Overall, I would recommend the authors to review the current version and carefully see what is the exact reduction to ERM that the authors are studying. If the loss function is not fixed, the studied analysis approach cannot be directly used and more caution is required. In my opinion, a Bellman residual error minimization setting (Antos et al, 2008) will be more relevant here and can be studied with the two phase update scheme in Algorithm 1. In the current version, the setting does not seem plausible for a reduction to ERM and I'm therefore not sure of the correctness of the results/claims.

**Specific comments**:
1. A more thorough and detailed literature review is required. As mentioned, generalization and overfitting of policy evaluation is actually well studied in literature starting from the finite sample based fitted value iteration schemes to FQE, DICE style methods in the literature.
2. In the references section, the citations are not proper as complete reference to publication venue and details are not included.
3. "No realizability assumptions on the ability of function class to fit values are needed": This is a wrong interpretation as the accuracy of any fitted evaluation procedure *will depend* on the approximation power of the value function class. Using a stability based analysis is helpful because it allows one to circumvent the uniform convergence argument and no assumption on the structural complexity of the function class are required.
4. In the first two pages, partial fitting is mentioned without any explanation, and therefore, it is hard to understand what it actually means.
5. Section 3, pg 2: "distribution over actions: \pi": \pi instead of pi.
6. Section 3.1: Significant updates required. Define MSBE, discuss how this is equivalent to ERM and how it connects to the stability analysis in Hardt et al.
7. Algorithm 1: line 7,10: t index is missing.
8. Paragraph right after algorithm 1: "...s. This prevents the step sizes from growing too quick..." $\alpha$ is step size not the whole update?
9. Instead of fitted expected SARSA, refer to this as FQE as used in previous works.
10. Section 3.2: More context is needed here as to how Algorithm 1 connects to this. Simply using a sample based minimization doesn't mean that it directly reduces to the same setting.
11. Assumption 2: It is one of the strongest assumptions used in the literature as it directly makes the assumption on a per-state basis. Refer to Chen and Jiang 2019 for more details.
12. Section 4.2: What is smoothness here for the linear function approximators?
13. Theorem 1: Whatever generalization the result implies, the implications of the value of k seem unclear. Is it correct to notice that keeping $K=1$ is the best possible bound? If true, what is the bias-variance tradeoff?
14. Again, in this section, various references to structure of the optimization problems are made, but the optimization problem itself is unclear.
15. In appendix, what does the subscript $X_{0:k}$ means at different places? The steps don't seem to use $k$ anywhere in certain proofs.
16. A full and detailed proof for Theorem 1 would have cleared the paper for both the authors and the reader. Flushing out the whole detailed proof will be helpful.


**References**
Le, Hoang, Cameron Voloshin, and Yisong Yue. "Batch policy learning under constraints." International Conference on Machine Learning. PMLR, 2019.

**Summary Of The Paper:**

The paper aims to study the effect of stability on generalization of a particular off-policy policy evaluation algorithm. Specifically, the authors study a version of Fitted Q-evaluation where the iterative procedure is instead thought of as a gradient update and is performed by only partially fitting the new Q-function to the current target Q function. The authors look at a linear function approximation setting and connect the recent results on stability of stochastic gradient methods (Hardt et al, 2016) with their method and show stability results, and connect that with overfitting of the policy evaluation scheme. The authors discuss the effect of the number of updates performed in each epoch during the partially fitted linear-update scheme and further show empirical simulations.

**Summary Of The Review:**

The paper attempts to connect an iterative fitted policy evaluation method with stability based generalization bound analysis of stochastic minimization algorithms. However, the reduction to a fixed risk measure for the partially fitted procedure is not discussed and is likely not applicable in this setting. Hence, the analysis, claims and connections of the proposed algorithm and stability of ERM are not clear in this paper. Further, various key quantities are not defined in the paper and therefore are not adequately discussed in the formal analysis. As such, I feel that this paper is not ready for publication and needs a significant and detailed revision.

---

> ### Author Response · Authors · 2021-11-23
> **Review Response**
>
> Thank you for taking the effort to provide such a constructive review. We agree that Residual Bellman Minimisation, and the associated formalism are a better setting for this work, and will take great care to introduce this in subsequent versions of the paper. We'd like to say that our proofs are already suited to BRM as discussed in Antos (2008), though more work needs to be done to establish this claim. $\hat G(w)$ does come into our proof, through the phrase: "The use of the supremum norm is unconventional here. We do this because under the supremum norm,the target network update is nonexpansive", though we agree we could be more clear about this. We will make sure to include more discussion of the specifics of our loss function,  optimisation procedure, and connection to the supervised learning setting, which will clarify some of the details that are murky here. Thanks once more for the advice.

---

### Official Review · Reviewer_ZmZV · 2021-11-03

**Correctness:** 3
**Technical Novelty And Significance:** 3
**Empirical Novelty And Significance:** Not applicable
**Recommendation:** 3
**Confidence:** 2

**Main Review:**

Strengths:
The assumptions, lemmas and theorems are clearly stated. The target algorithm to analyze is clear from the pseudo code. The paper is clearly written.

It attempts to analyze the "generalization" properties of a batch RL learning target through a definition of stability, which is novel.

Weaknesses:
The main weakness of the paper is its usefulness to batch RL learning. It would be nice for the author to devote more time on clarifying how resting on many assumptions and viewed from a particular angle of stability, Theorem 1 actually can help us understand general batch RL learning better, or help us devise a better batch RL algorithm.

One part I didn't understand is that "off-distributional stability" from the paper seems to refer to stability in regard to data shuffling, whereas the term "generalization" in RL literature usually refers to how the model learned from a training distribution performs on a possibly different test distribution.

It would also be nice to see a discussion of the learning target of choice and particularly why using a target network is chosen for this analysis, instead of for example, using semi-gradient or using other off-policy learning algorithms altogether.

Nits:
1. first line : "wort-case bounds"
2. Figure 1 is really hard to read.


**Summary Of The Paper:**

The paper analyzed a case of the stability property due to data shuffling for an expected SARSA update using linear function approximation and target networks.

**Summary Of The Review:**

I think the paper makes an interesting theoretical contribution in a limited setting. It may need a bit more work to bring the message home and make it significantly meaningful and illuminating in designing new algorithms to practitioners in batch RL.

---

> ### Author Response · Authors · 2021-11-23
> **Review Response**
>
> Thanks so much for taking the time to review our work. We appreciate your confirmation of the novelty of this line of inquiry. While we feel that our work is mostly theoretical, and provides some practical insight into the effects of target networks, we agree that this relationship can be explored further, and that more ties to RL practice would make the work stronger. In future versions of the work, we will be certain to spend more effort characterising the specifics of the learning objective, as this was a common point of criticism across reviews. Thank you.

---

### Author Response · Authors · 2021-11-23
**Response to Reviews**

We'd like to thank all the reviewers for their time and care. From the reviews, it's clear to us that everyone agrees that the work is meaningful, which is encouraging to read, despite the fact that more attention to detail is needed to bring it to a point where it is ready to publish. Specific comments are included as replies to individual reviews. We're grateful for all the advice, and are hopeful that the inclusion of everyone's feedback will make this a solid paper.

---

### Decision · Program_Chairs · 2022-01-20

**Decision:**

Reject

**Comment:**

In this paper, the authors studied algorithmic stability of batch reinforcement learning algorithms, as well as its connection to certain generalization bounds (motivated by the prior work Hardt et al developed for SGD on nonconvex optimization problems). While understanding the stability and generalization of batch RL is certainly an interesting and important direction, the paper in its current form is not yet ready to be published. As the reviewers pointed out, both the analyses and the claims need to be polished (in fact, important details and definitions are missing); and the theoretical contributions are only made in a limited setting.